# Outlier Denoising Using a Novel Statistics-Based Mask Strategy for Compressive Sensing

**Weiqi Wang** [1] 🆔**, Jidong Yang** [1,]*🆔**, Jianping Huang** [1]**, Zhenchun Li** [1] **and Miaomiao Sun** [2]

1    Key Laboratory of Deep Oil and Gas, School of Geosciences, China University of Petroleum (East China), Qingdao 266580, China
2    Software & Service Outsourcing/IT, Qingdao Vocational and Technical College, Qingdao 266555, China
*    Correspondence: jidong.yang@upc.edu.cn

**Abstract:** Denoising is always an important step in seismic processing, in order to produce high-quality data for subsequent imaging and inversion. Different types of noise can be suppressed using targeted denoising methods. For outlier noise with singular amplitudes, many classical denoising methods suffer from signal leakage. To mitigate this issue, we developed a statistics-based mask method and incorporated it into the compressive sensing (CS) framework, in order to remove outlier noise. A statistical analysis for seismic data amplitudes was first used to identify the locations of traces containing outlier noise. Then, the outlier trace locations were compared with a mask matrix generated by jitter sampling, and we replaced the sampled traces of the jitter mask that had the outlier noise with their nearby unsampled traces. The optimized sampling matrix enabled us to effectively identify and remove outliers. This optimized mask strategy converts an outlier denoising problem into a data reconstruction problem. Finally, a sparsely constrained inverse problem was solved using a soft-threshold iteration solver to recover signals at the null locations. The feasibility and adaptability of the proposed method were demonstrated through numerical experiments for synthetic and field data. The results showed that the proposed method outperformed the conventional f-x deconvolution and median filter method, and could accurately suppress outlier noise and recover missed expected signals.

**Keywords:** outlier noise; denoising; compressive sensing; inverse problem

## 1. Introduction

In exploration seismology, the primary reflection is usually considered as the effective signal, while the others are regarded as noise [1]. Various noises reduce the signal-to-noise ratio (SNR) of prestack data, and degrade the quality of seismic imaging and inversion results. Improving the SNR of seismic data aims to suppress unwanted signals without scarifying the quality of effective signals, which is an important step in seismic data preprocessing [2]. With rapid developments in seismic acquisition and increasing complexity of near-surface condition and subsurface exploration targets, improving noise removal is a long-standing hot topic.

According to amplitude, frequency, and propagation characteristics, noise can be categorized into regular and irregular groups. Regular noise has a specific frequency range and apparent velocity. A representative signal is the surface wave that propagates near the surface. Irregular noise occurs randomly, and has no typical frequency band and propagation direction. To improve the SNR of seismic data, many methods have been proposed, such as median filters [3,4], predictive filters [5–8], sparse transformation [9–12], rank reduction theory [13,14], empirical mode decomposition [15–20], and deep learning assistant methods [21–25]. Although a lot of tests demonstrate that these methods can effectively suppress random and linear noise, they are difficult to completely remove outlier noise.

Outlier noise usually refers to one or a group of waveforms that are significantly different from the effective signals in terms of amplitudes [26]. Outlier detection has been widely adopted in many fields, including medical diagnosis, weather forecasting, intrusion detection systems, and fraud control. In seismic data processing, the occurrence of outliers leads to large errors in the overall statistics of the data, especially in the prediction filter and least squares denoising [27]. Since the characteristics of seismic data are commonly different from pictures, image-related signal processing methods cannot be directly applied to seismic data. Thus, according to the Z-Score and modified Z-score in traditional image processing, it is difficult to effectively suppress the outlier noise of seismic data. In seismic data processing, median filtering has been used to suppress outlier noise, but signal leakage is inevitable. Gemmeke et al. [28] used an over-complete dictionary of clean speech samples to find the sparsest sample combination that jointly approximates the non-missing features of noisy signals, and replaced missing samples with linear combinations of clean speech samples. Gholami and Sacchi [29] proposed two sparsity-based deconvolution methods in the time and frequency domains where the outliers are ignored, while generating solutions with isolated spikes. Zhao et al. [30] designed a filter that jointly minimizes the $L_2$-norm of the random component and the $L_1$-norm of the irregular noise. The computational complexity of sparse-domain filtering is several times larger than that of traditional sparse threshold methods. Jeong et al. [31] proposed a local outlier detecting factor (LoOF), which can separate the source interference noise in a homologous acquisition. Chen et al. [32] proposed a robust residual dictionary learning method to retrieve reflection signals from leaked signals.

The above methods have been proven to effectively suppress outlier noise. However, there are still some issues in terms of the computational efficiency and adaptability for complex low-SNR seismic data. In this study, we proposed a novel statistic-based mask strategy, and incorporated it into the compressed sensing (CS) framework to remove outlier noise, which transformed the outlier denoising problem into a data reconstruction problem. Traditional compressed sensing denoising methods show good performance for suppressing random noise; however, as a least squares-based sparse inversion algorithm, CS is particularly sensitive to outlier noise, and may even introduce more noise for nearby traces. First, by applying a statistical analysis for the amplitude standard deviation, the traces with outlier noise are identified and used to design a mask to zero noisy traces. Then, the outlier trace locations are compared with the mask matrix generated by jitter sampling, which replaces the sampled traces of the jitter mask that had the outlier noise with its nearby unsampled traces. Finally, the missing data at the null trace are reconstructed with a curvelet-domain iterative threshold algorithm, which can suppress both outlier and random noise. Numerical examples for both synthetic and field data demonstrated the feasibility and adaptability of the proposed method.

## 2. Methods

### 2.1. Review of CS Denoising Theory

Seismic data denoising removes unwanted signals and enhances the expected signals. Noisy seismic data can be expressed as follows:

$$d = s + n \qquad (1)$$

where $d$ denotes the observed data, and $s$ and $n$ denote effective signal and noise, respectively. By introducing a denoising operator $N$, the denoising process can be expressed as follows:

$$s_0 = Nd \qquad (2)$$

$$n_0 = d - Nd \qquad (3)$$

where $s_0$ and $n_0$ represent the signal and noise after denoising, respectively. In the ideal case, $s_0 = s$, $n_0 = n$, but this is not always the case in real applications. In most cases, the

signal leakage caused by denoising is difficult to eliminate, especially for the data with outlier noise.

Donoho et al. (2006) [33] proposed the CS theory, which transforms traditional signal acquisition into information acquisition of data. Based on the sparse characteristics of the signal and non-sparsity of random noise, the effective signal can be accurately separated and extracted from noisy data during the reconstruction process. A CS framework can be summarized as follows:

$$d = \Theta x = \Phi \Psi x \tag{4}$$

where $\Theta \in \mathbf{R}^{n \times N}$ represents the sensing matrix, n is the number of total data points, and $N$ is the number of sampled data points ($N << n$). $\Phi \in \mathbf{R}^{n \times N}$ and $\Psi \in \mathbf{R}^N$ represent the observation and sparse matrices, respectively. The sparse coefficient $x \in \mathbf{R}^N$ is obtained by projecting the data onto the sensing matrix. Based on the CS framework, seismic data reconstruction can be implemented using a sparse inversion algorithm as follows:

$$\hat{x} = \arg\min \|x\|_0 \quad s.t \quad d = \Phi \Psi x \tag{5}$$

where $x$ is the estimated value. The underdetermined Equation (4) is an $L_0$-norm minimization problem, and it is usually difficult to solve. Candes and Tao [34] solved this problem by limiting the sensing matrix $\Theta$ to satisfy the restricted isometric property (RIP), transforming the $L_0$-norm into an easy-to-solve $L_1$-norm convex optimization problem. The RIP criterion can be written as follows:

$$(1 - \delta)\|x\|_2^2 \le \|\Theta x\|_2^2 \le (1 + \delta)\|x\|_2^2 \tag{6}$$

where $\delta$ ($0 < \delta < 1$) represents the restricted isometric coefficient. The matrix that satisfies the RIP property can be used as a sensing matrix for CS reconstruction. However, in reality, it is difficult to determine whether the matrix satisfies the RIP criterion. Baraniuk [35] proved the equivalence of RIP and the uncorrelation between the measurement and sparse matrices, which can be mathematically expressed as shown below:

$$\mu(\Phi, \Psi) = \sqrt{n} \max_{1 \le i,j \le n} \left| \langle \phi_i, \psi_j \rangle \right| \tag{7}$$

where $\mu(\Phi, \Psi) \in [1, \sqrt{n}]$ represents the coherence degree of the measurement and sparse matrices. The small $\mu$ indicates a low correlation, and vice versa. Therefore, Equation (5) can be rewritten as an $L_1$-norm optimization problem:

$$\hat{x} = \arg\min \|x\|_1 \quad s.t \quad d = \Phi \Psi x \tag{8}$$

Herrmann and Hennenfent [36] proposed the curvelet-based sparsity-promoting inversion (CRSI) method to solve Equation (8). Here, we choose the curvelet basis functions to construct a sparse matrix $\Psi$, which is suitable for seismic data processing. It can provide an optimal sparse representation of signals with piecewise smooth curve edges, and has multi-scale, multi-direction, and localization characteristics. The curvelet basis functions are shown in Figure 1. The left side represents the spatial scale viewpoint, the middle represents the angle coefficient of the frequency components, and the right side is the curvelet coefficient viewpoint.

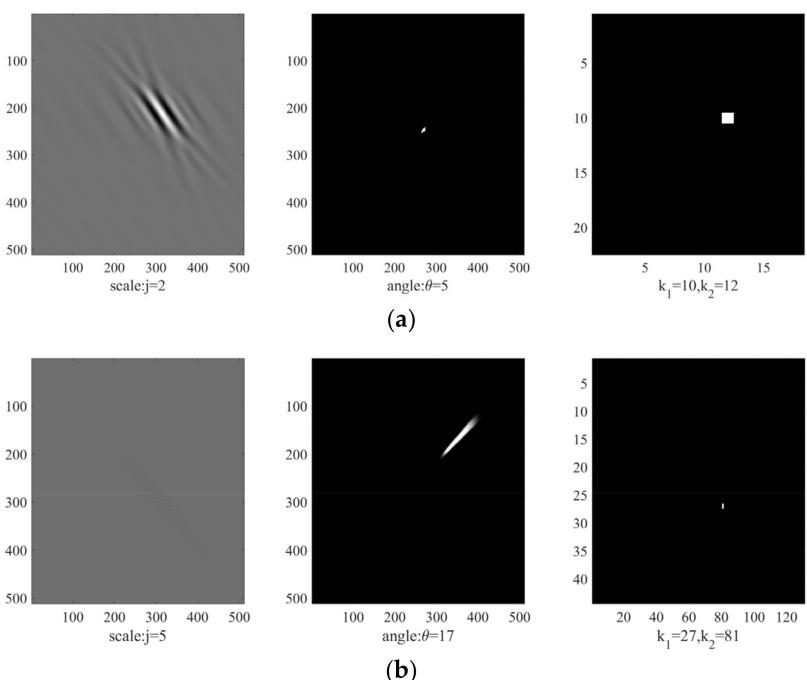

**Figure 1.** Diagram curvelet-based function. (**a**) The coarse scale. The left side represents the spatial scale viewpoint (j = 2), the middle represents the angle coefficient ($\theta$ = 5) of the frequency components, and the right side is the curvelet coefficient ($k_1$ = 10, $k_2$ = 12) viewpoint. (**b**) The fine scale. The left side represents the spatial scale viewpoint (j = 5), the middle represents the angle coefficient ($\theta$ = 17) of the frequency components, and the right side is the curvelet coefficient ($k_1$ = 27, $k_2$ = 81) viewpoint.

The jittered undersampling matrix is selected as the measurement matrix $\boldsymbol{\Phi}$ [37], which guarantees randomness and controls the maximum sampling gap size. The jittered-sampled data points are given by the following:

$$\begin{aligned}
&\boldsymbol{d}[i] = \boldsymbol{f}[j] \quad for \ i = 1, \dots, n \ and \\
&j = \tfrac{1-\gamma}{2} + \gamma \cdot i + \varepsilon_i
\end{aligned} \tag{9}$$

where $\gamma$ ($\gamma$ = 1, 3, 5, ...) represents an undersampling factor; $\varepsilon_i$ denote discrete random variables, which are identically distributed according to a uniform distribution on the interval between $-\lfloor (\xi - 1)/2 \rfloor$ and $\lfloor (\xi - 1)/2 \rfloor$. $\xi$ ($0 \le \xi \le \gamma$) represents the jitter parameter.

### 2.2. A Novel Statistics-Based Mask Function for Outlier Noise

The construction principle of the traditional CS measurement matrix only limits randomness and maximum gap. If seismic data have outlier noise, the outlier traces will affect their nearby reconstruction results and produce serious signal leakage. Therefore, an optimized mask function should be designed to reduce the effects of outlier noise in the application of CS denoising.

Based on the statistical analysis, we chose the standard deviation $\sigma$ to identify the outlier traces. For the $j$th trace, the standard $\sigma_j$ can be computed as follows:

$$\sigma_j = \sqrt{\frac{\sum_{i=1}^n (x_i - \mu)^2}{n}} \tag{10}$$

where $x_i$ represents the atom on the $j$th trace, $\mu$ represents the amplitude mean on the $j$th trace, and $n$ is the total number of atoms on the $j$th trace.

In addition, we also computed a standard deviation σ for all traces to compare with the single-trace standard deviation $\sigma_j$, in order to calculate an outlier mask factor $T_o$ as follows:

$$T_o = \frac{\sigma_j}{\sigma} \tag{11}$$

According to the magnitude of $T_o$, we identified the location of traces that contain outlier noise. When the outlier mask factor is 1, it means that the amplitude energy of the single trace is consistent with the energy of the data. Where $T_o = 0$, this denotes that the trace is null, and $T_o = 1$ denotes that the trace energy is equal to the energy of the data. For $T_o > 1$, it indicates that the amplitude energy of the trace exceeds the normal level. For $0 < T_o < 1$, it indicates that the trace is a weak signal. From many numerical experiments for synthetic and field data, we found that the range of $T_o$ is from 0.2 to 3 for effective data with regular amplitudes. Based on these observations, the statistics-based outlier mask matrix $\boldsymbol{\Phi}_t$ can be constructed as follows:

$$\boldsymbol{\Phi}_t(x_e) = \begin{cases} 0 \overset{def}{=} \text{normal}, & 0.2 < T_o < 3 \\ 1 \overset{def}{=} \text{outlier}, & \text{otherwise} \end{cases} \tag{12}$$

Equation (12) refers to when the $T_o$ value of a trace is outside the selected range; this means that the trace is an outlier. $\boldsymbol{\Phi}_t$ is a mask matrix containing only 0 and 1, and 1 represents the location of the outlier trace in the noisy data.

When the CS measurement matrix is jitter undersampled, we can add the outlier mask to the sampling matrix. First, initialize the outlier mask $\boldsymbol{\Phi}_t(x_e)$ and jitter mask $\boldsymbol{\Phi}_m(y_e)$, where $x$ and $y$ represent the outlier trace and the zero-trace number of jitter sampling, respectively. If the trace satisfies $x_e = 1$ and $y_e = 0$, it continues to judge other positions without making changes. When $x_e = 1$ and $y_e = 1$, it means that the data of $e$ trace is an outlier trace, and it is sampled to the jitter matrix now, the current position is judged $\eta = \min(|\boldsymbol{\Phi}_t(x_e) - \boldsymbol{\Phi}_m(y_e)|)$, where $\eta$ represents the distance outlier trace number in the jitter matrix nearest to the location of sampling. Then, we exchange its $\boldsymbol{\Phi}_m(y_\eta)$ and $\boldsymbol{\Phi}_m(y_e)$ values, in order to ensure that the newly constructed jittered undersampling matrix $\boldsymbol{\Phi}_{tm}$ does not contain the outlier trace. Finally, $\boldsymbol{d}_s = <\boldsymbol{\Phi}_{tm}, \boldsymbol{d}>$ represents removing the outlier traces of noisy data. The above process can be clearly shown by Figure 2.

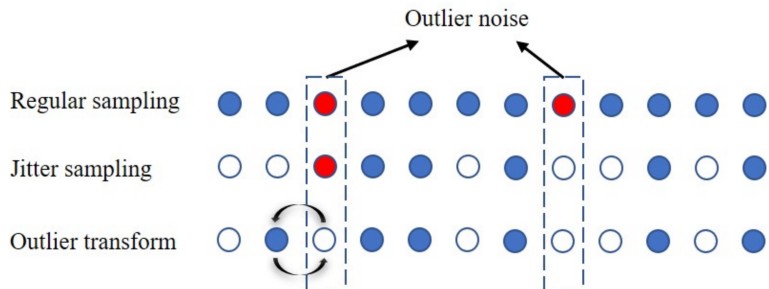

**Figure 2.** The outlier mask replacement strategy. The red dots represent outlier traces, the blue dots represent sampling traces, and the hollow dots represent unsampled locations.

### 2.3. CS Denoising Algorithm with the Proposed Mask

Based on the above method, the problem of denoising data with outlier noise can be converted into a reconstruction problem of $\boldsymbol{d}_s$ data without outlier noise. Equation (8) can be converted into the following constrained optimization problem:

$$P^\lambda = \begin{cases} \hat{\boldsymbol{x}}_\lambda = \arg\min_x \frac{1}{2}\|\boldsymbol{d} - \boldsymbol{\Phi}_{tm}\boldsymbol{\Psi}\boldsymbol{x}\|_2^2 + \lambda\|\boldsymbol{x}\|_1 \\ \boldsymbol{d}_\lambda = \boldsymbol{\Psi}\boldsymbol{x}_\lambda \end{cases} \tag{13}$$

where $\boldsymbol{\Phi}_{tm}$ represents the mask matrix without outlier traces; $\lambda$ represents the Lagrange multiplier, which determines the proportion of the $L_2$ error term and the $L_1$-norm term. Thus, solving the $P$ problem is transformed into continuously reducing $\lambda_\varepsilon$ and adjusting $\lambda$ to solve the $P^\lambda$ problem, where $\lambda_\varepsilon = sup_\lambda \left\{ \lambda : \|\boldsymbol{d} - \boldsymbol{\Phi}_{tm}\boldsymbol{\Psi}\hat{x}_\lambda\|_2^2 \le \varepsilon \right\}$. The exact solution is approached by increasing the energy information in the data by continuously decreasing $\lambda$.

In this study, we used the gradient descent method to solve Equation (13). In the iterative process, the threshold parameter $T_\lambda$ according to the curvelet basis of each scale was continuously reduced until an approximate solution was finally obtained. The corresponding algorithm is summarized in Algorithm 1.

---

**Algorithm 1** CS denoise based on the CRSI method

---

Input: measurement matrix $\boldsymbol{\Phi}_{tm}$, measurement data $\boldsymbol{d}$,
Output: signal estimation $\hat{\boldsymbol{x}}$

(1)　Initialize: Curvelet threshold parameter $\lambda_i = \|\boldsymbol{\Psi}x\|_\infty > \lambda_1 > \lambda_2 > \lambda_3 > \ldots$, the maximum iteration of the outside loop $L_{out}$, the maximum iteration of the outside loop $L_{out}$, residual $\varepsilon = 0$, $k = 0$, $j = 0$, $\hat{x}^0 = 0$;

(2)　Iteration start:
 if $\left\|\boldsymbol{d} - \boldsymbol{\Phi}_{tm}\boldsymbol{\Psi}x^k_{\lambda_i}\right\|_2 > \varepsilon \text{ and } k \le L_{out}$
 　　for j = 0: $L_{\text{in}-1}$
 $x^{j+1}_i = T_{\lambda_{ik}}\left(x^j_i + \boldsymbol{\Theta}^T_i(\boldsymbol{d} - \boldsymbol{\Phi}_{tm}\boldsymbol{\Psi}_i x^j_{\lambda_i})\right);$
 　　end
 $x^k_i = x^{j+1}_i;$
 end

(3)　Output: $\hat{\boldsymbol{x}}$

---

For low SNR data, the soft threshold function can effectively avoid sudden changes in the process of threshold removal, avoiding local jitter in the results after denoising. The soft threshold is smoothed in the processing of curvelet coefficients. Thus, $T_\lambda(\boldsymbol{x}_i)$ chooses the soft threshold function, which gradually decreases with an increase in the number of iterations to approach the exact solution, as follows:

$$T_{\lambda_{ik}}(\boldsymbol{x}_i) := \text{sgn}(x_i) \cdot \max(0, |\boldsymbol{x}_i| - |\lambda_{ik}|) \tag{14}$$

After the curvelet coefficient is finally obtained, the reconstructed data $\hat{d}$ follow in Equation (15):

$$\hat{\boldsymbol{d}} = \boldsymbol{\Psi}^T\hat{\boldsymbol{x}} \tag{15}$$

here, $\boldsymbol{\Psi}^T$ represents the inverse curvelet transform. The reconstructed data $\hat{d}$ without outliers and random noise were obtained.

## 3. Results

In this section, we test the proposed method using a synthetic model and two field datasets. We used the SNR and peak signal-to-noise ratio (PSNR) as a criterion for evaluating the denoising effect, which is given by the following:

$$\text{SNR} = 10\log_{10}\left(\frac{\|\boldsymbol{x}\|_2^2}{\|\hat{\boldsymbol{x}} - \boldsymbol{x}\|_2^2}\right) \tag{16}$$

$$\text{PSNR} = 10\log_{10}\left(\frac{max(\boldsymbol{x})^2}{\text{MSE}}\right) \tag{17}$$

where MSE denotes the mean square error and is calculated as shown below:

$$\text{MSE} = \frac{1}{mn}\sum\nolimits_{i=0}^{m-1}\sum\nolimits_{j=0}^{n-1}(\boldsymbol{x} - \hat{\boldsymbol{x}})^2 \tag{18}$$

### 3.1. Synthetic Data

The noise-free synthetic data are shown in Figure 3a,b, with the data of random and outlier noise. The SNR of the noisy data was −5.25 dB, and the PSNR was 12.47 dB. We first applied the proposed method for noisy data to construct the outlier mask matrix. Figure 4a,b show the two kinds of standard deviation distributions of the data with outlier noise, and the threshold was calculated by $T_o$ proposed previously. There was an apparent difference between outlier noise and effective signals. For this dataset, $1 \leq T_o \leq 3$ was a good range to distinguish outlier traces. We designed the outlier mask matrix based on the $T_o$ value, and used a combination of the strategy and the jitter matrix in this research, as shown in the Figure 5a matrix. The mask matrix was then used to interpolate the data with outlier noise (Figure 5b). Figure 6 shows the denoising results using a traditional median filter and the proposed method. Figure 6a,b show the median filtering results and the corresponding noise. The filter length of the median filter was three. Although the outlier and random noise were removed, there was strong signal leakage in the noise (Figure 6b). The missed data of the outlier traces removed by the mask matrix was reconstructed by CS (Figure 6c). The SNR of the reconstructed results was 8.42 dB, and the PSNR was 25.72 dB. Random and outlier noise were well suppressed, and the difference showed that reflection signal were not leaked (Figure 6d). From single-trace analysis (Figure 7), we see that the signals were not destroyed before and after denoising, and that the amplitudes became normal after removing the outliers. Based on the median filtering and the proposed method, a stability test of multiple SNRs and PSNRs was carried out (Figure 8). From the results, it can be seen that as the noise increased, the denoising effect also decreased, but the proposed method still had great advantages over traditional methods.

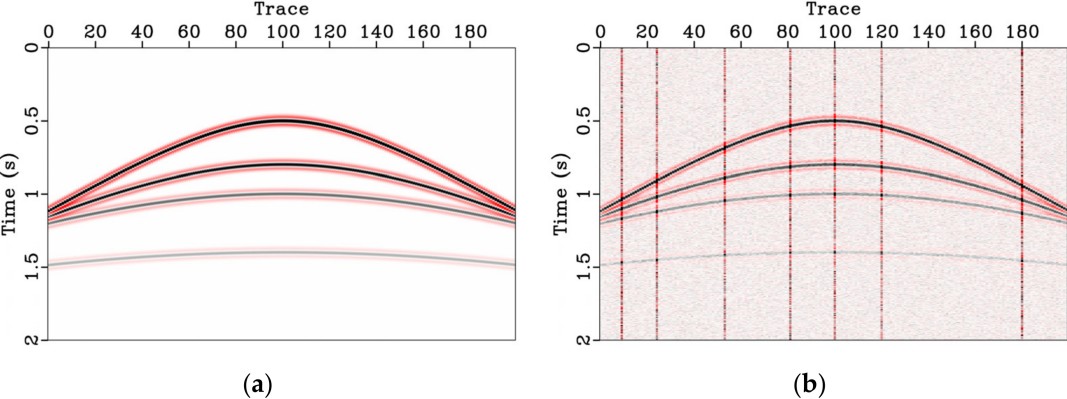

(**a**)                                    (**b**)

**Figure 3.** Synthetic data. (**a**) A noise-free common-shot gather. (**b**) Noisy gather with random and outlier noise (SNR = −5.25 dB, PSNR = 12.47 dB).

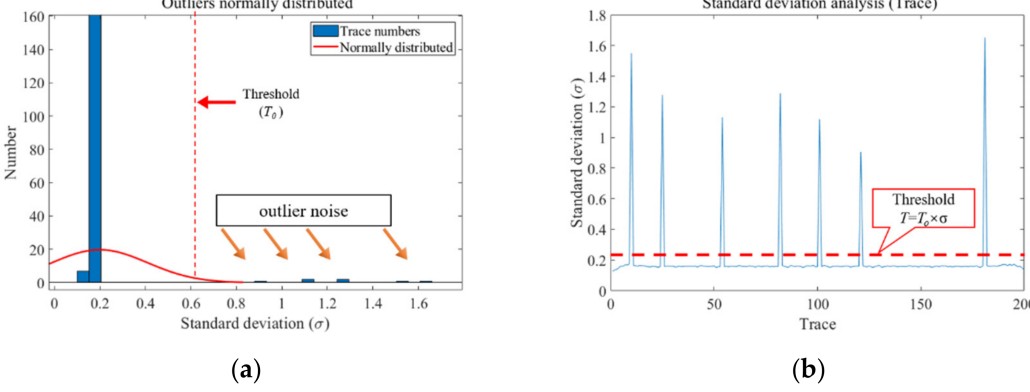

(**a**)                                    (**b**)

**Figure 4.** The statistical results of outlier noise for the data in Figure 3b. (**a**) Energy distributed analysis. (**b**) Threshold processing diagram.

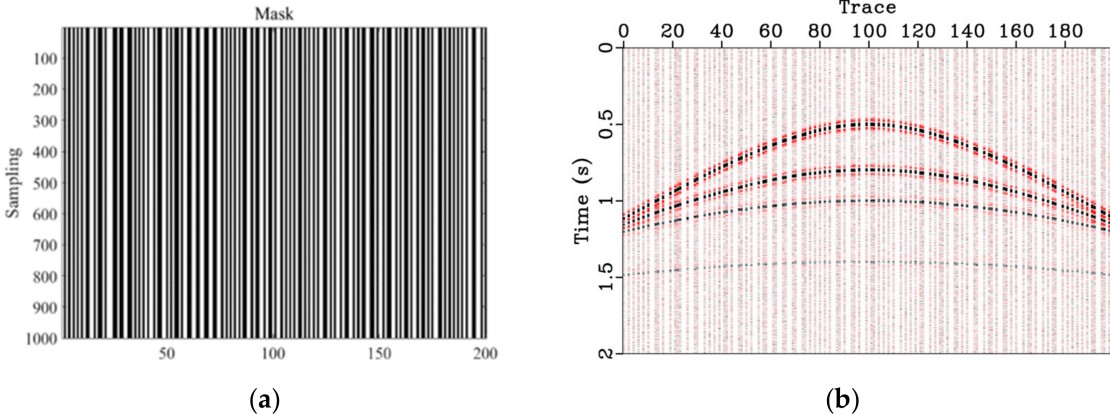

**Figure 5.** Mask jitter matrix. (**a**) The 50% jittered undersampling matrix (removed outlier trace). (**b**) The decimated synthetic model with 50% removed traces (with outlier noise).

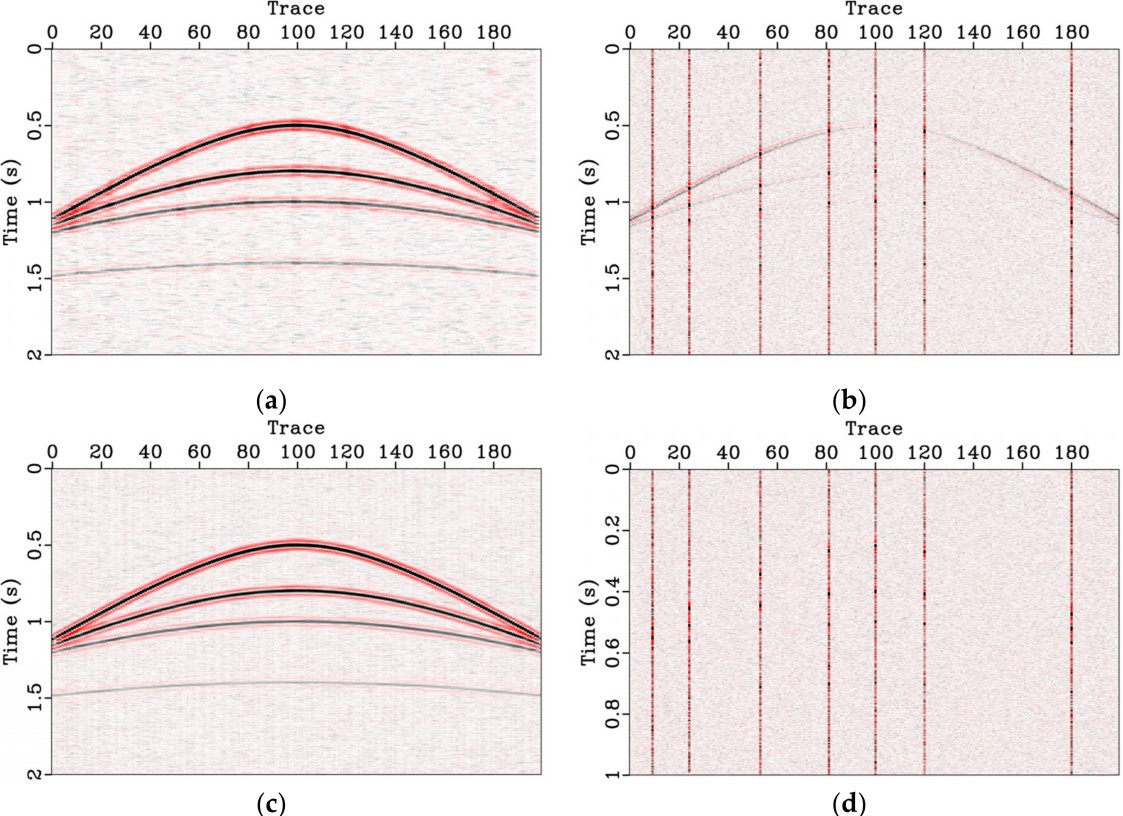

**Figure 6.** A comparison of denoising performance. (**a**) The denoised results of median filtering (SNR = 6.56 dB, PSNR = 17.37 dB). (**b**) Removed outlier noise. (**c**) The denoised results of the proposed method (SNR = 8.48 dB, PSNR = 25.72 dB). (**d**) Removed outlier and random noise.

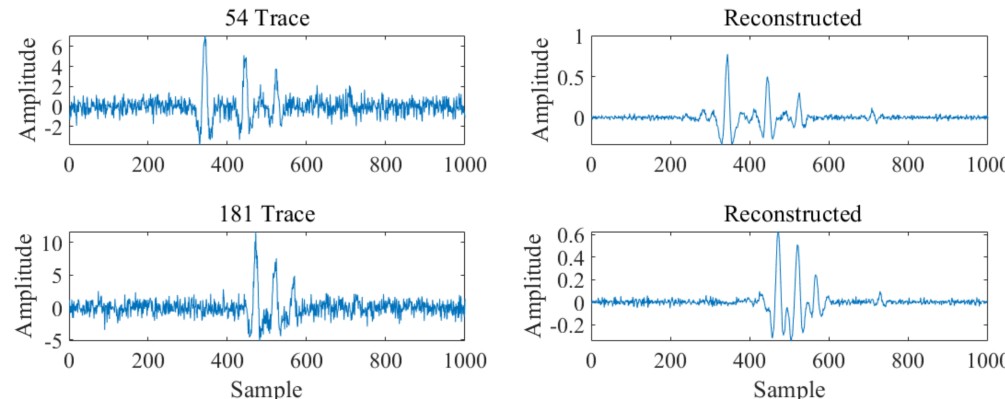

**Figure 7.** Single-trace contrast before and after outlier denoising.

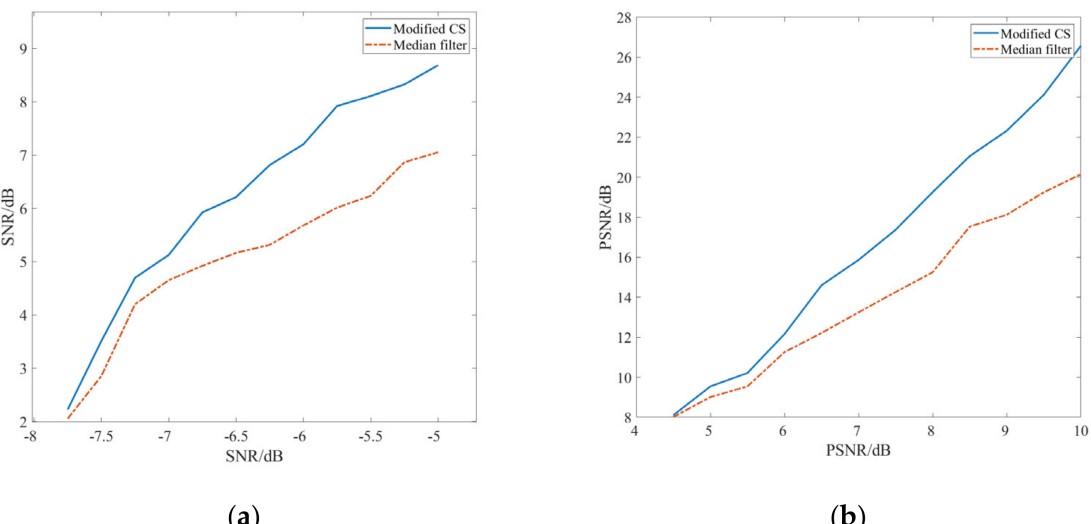

(**a**)  (**b**)

**Figure 8.** Comparison of denoising effect of different noise levels. (**a**) Multi-SNR levels denoising curves. (**b**) Multi-PSNR levels denoising curves.

### *3.2. Marine Data*

The second example involved marine data, which are shown in Figure 9. It had 180 traces and 500 sampling points, and the sampling rate was 4ms. Eight outlier traces were added to the original marine data to test the denoising performance of the proposed optimized outlier matrix. The SNR of the noisy data was −2.09 dB, and the PSNR was 9.32 dB. The standard deviation of the data was calculated as shown in Figure 10. Figure 11b show the noisy data. According to the outlier noise distribution, $1 \leq T_0 \leq 2$ was the best range to distinguish outlier traces. In order to protect the effective signal, we used the POCS algorithm to reconstruct the missing data (Figure 11b). The choice between iterative threshold algorithm and the POCS algorithm is explained in Section 4. Figure 11c,d show the median filtering results and the corresponding noise. The filter length of the median filter was three. There was still obvious signal leakage in the removed noise. Figure 11e shows the CS reconstruction results that used the outlier matrix as the sampling matrix. The SNR of the denoised result was 7.20 dB, and the PSNR was 20.61 dB. This showed that the proposed mask matrix could be used as a good sampling matrix for CS outlier denoising and data reconstruction.

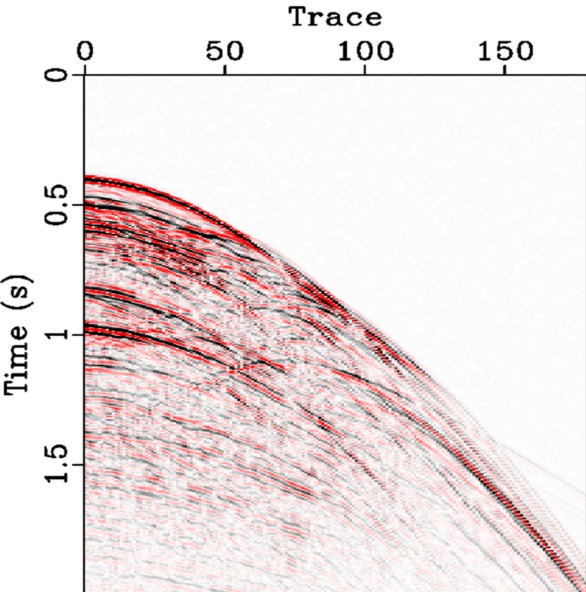

**Figure 9.** A common-shot gather of marine field data.

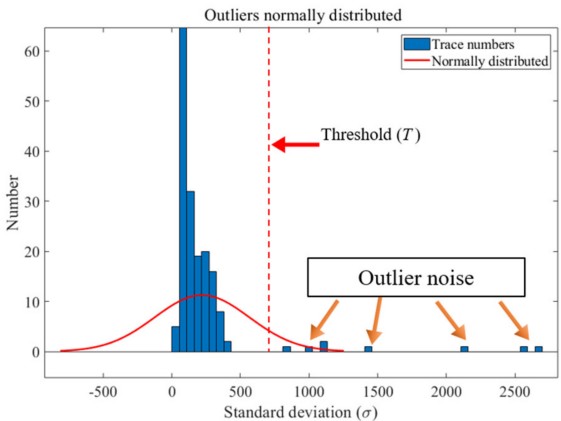

**Figure 10.** The statistical results of outlier noise for the noisy data.

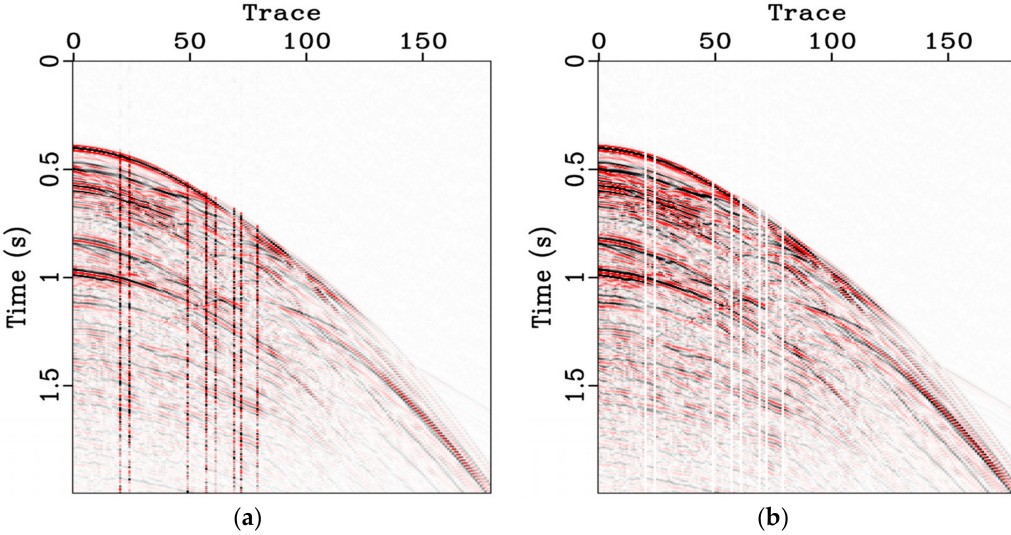

(a)          (b)

**Figure 11.** *Cont.*

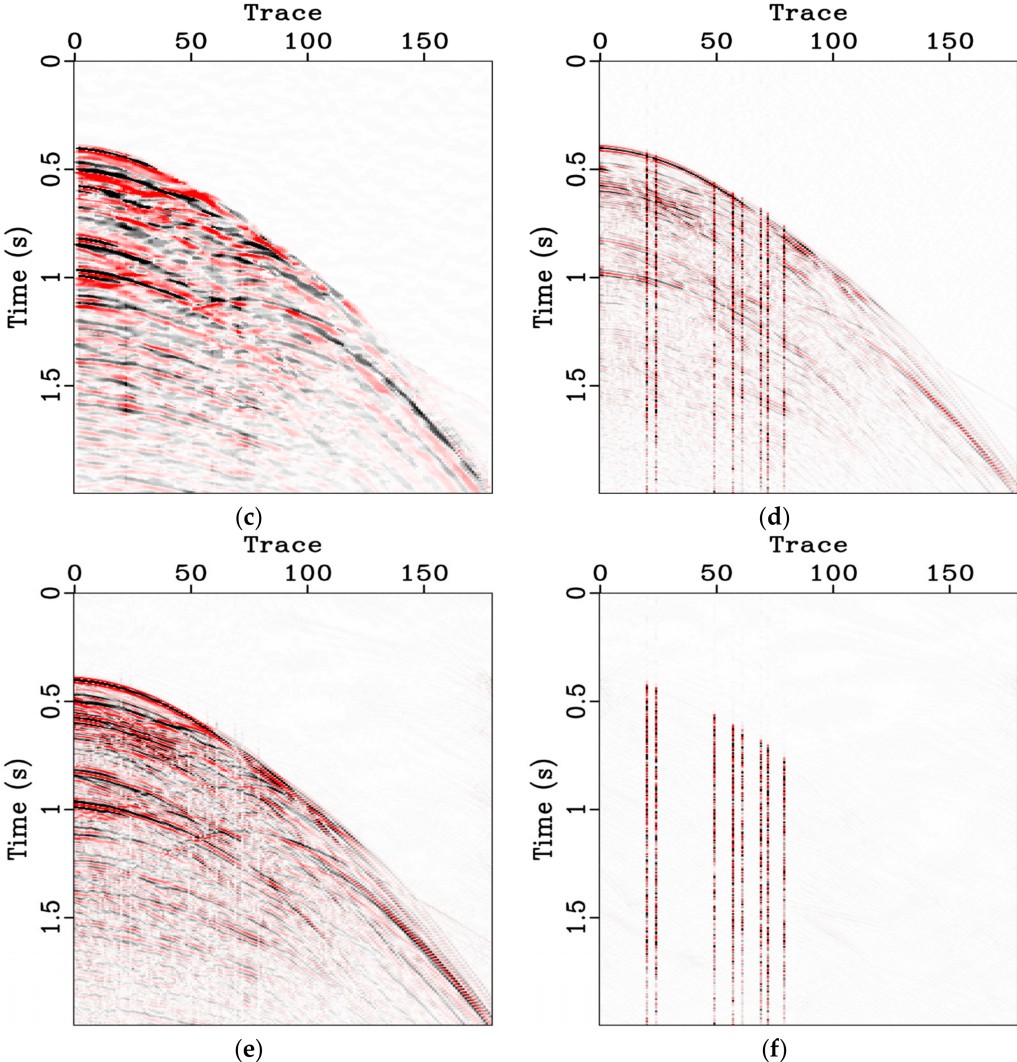

**Figure 11.** Marine data comparison of denoising performance. (**a**) Noisy data with outlier noise. (SNR = −2.09 dB; PSNR = 9.32 dB). (**b**) Removed traces with outlier noise. (**c**) The denoised results of median filtering (SNR = 6.14 dB, PSNR = 11.32 dB). (**d**) Removed noise by median filtering. (**e**) The reconstructed results of (**b**) (SNR = 7.20 dB; PSNR = 20.61 dB). (**f**) Removed noise by CS.

### 3.3. Land Field Data

The last example involved land field data, which had 3501 time samples and 701 traces. The time sampling was 2 ms. The data contained not only strong random and outlier noise, but also weakly reflected signals and null traces (Figure 12). Figure 13a shows the distribution of the standard deviation of trace amplitudes. Here, we identified not only outlier traces, but also weak signal and null traces, which only needed to take the lower bound of $T_o$ instead of 1. This meant that we needed to constrain the maximum and minimum values of $T_o$ separately. Finally, we selected $T_{omin}$ = 0.25 and $T_{omax}$ = 3 to distinguish outlier traces (Figure 13b). The outlier mask matrix $\Phi_t$ is shown in Figure 14a. From the analysis of the outlier mask matrix and the seismic data, it can be seen that the positions of the null traces, weak signal, and strong amplitude removed. Figure 14b represents the missing data without outlier noise. Figure 15a,c,e,g show conventional f-x deconvolution with a prediction filter with a length of ten sample points, CS iterative threshold denoising without considering the outlier traces (sampling rate 50%), the median filtered and f-x deconvolution (MF + FX) combined denoising method, and the proposed method denoising results, separately. Figure 15b,d,f,h represent the noise removed by different methods. The f-x deconvolution method was not stable for data with outlier noise,

and led to serious signal leakage. Traditional CS did not consider outlier noise, which led to a large error when calculating the threshold parameter of the curvelet domain. The MF + FX method suppresses the noise better. However, the signal could still be seen from the difference profile, random noise suppression was not complete, and the data energy after denoising was still unbalanced. The comparison shows that the CS method using the optimized sampling matrix can remove outlier noise while reducing signal leakage. In examining the black frame in Figure 12 more closely, the zoomed sections are shown in Figure 16 for better comparison. The continuity of the data improved significantly, and the random noise suppression was thorough. Figure 17 shows a single-trace (501th trace) comparison before and after denoising. The strong energy point masked the amplitude waveform of the normal signal. This shows the feasibility of the proposed method for land field data.

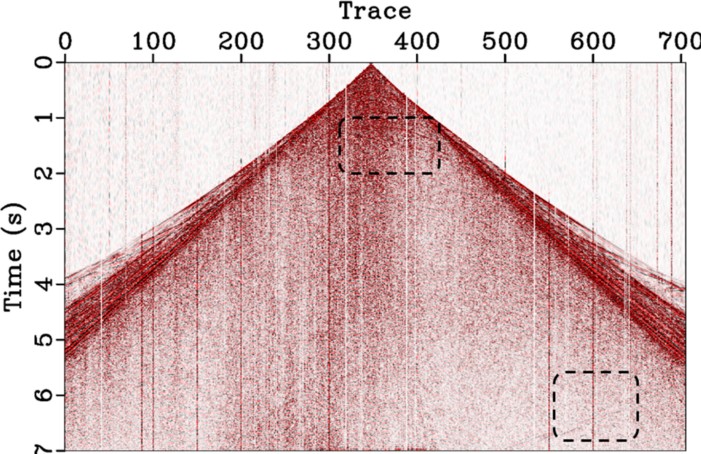

**Figure 12.** Land field data. Dashed box is follow-up the zoom contrast section.

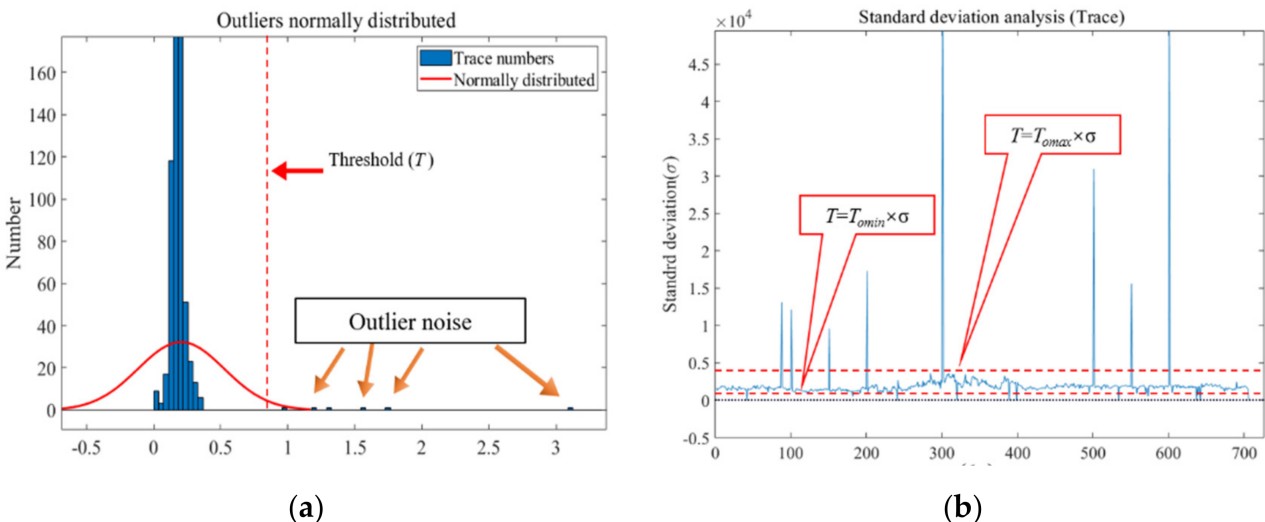

**Figure 13.** The statistical results of outlier noise for the noisy data. (**a**) Energy distributed analysis. (**b**) Threshold processing diagram.

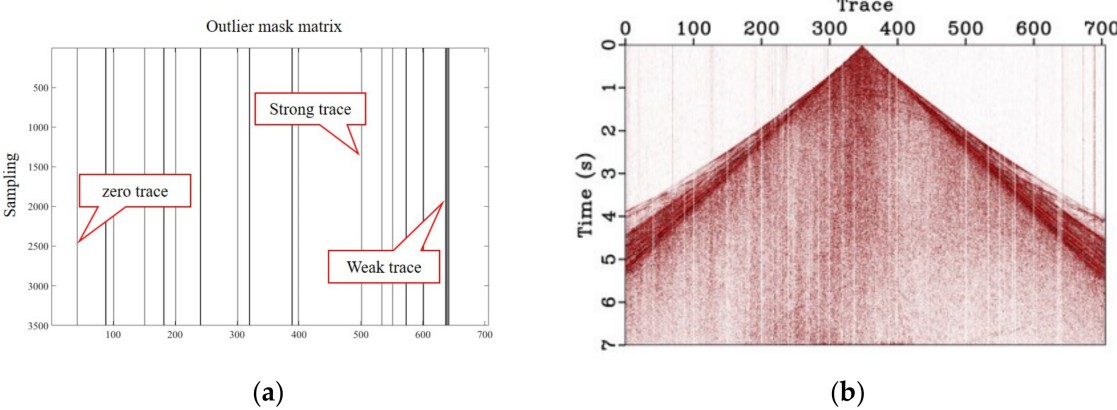

(a)

(b)

**Figure 14.** The removed outlier trace. (**a**) Outlier mask matrix ($\boldsymbol{\Phi}_t$). (**b**) Data without outlier noise ($\boldsymbol{d}_s$).

(a)

(b)

(c)

(d)

(e)

(f)

**Figure 15.** *Cont.*

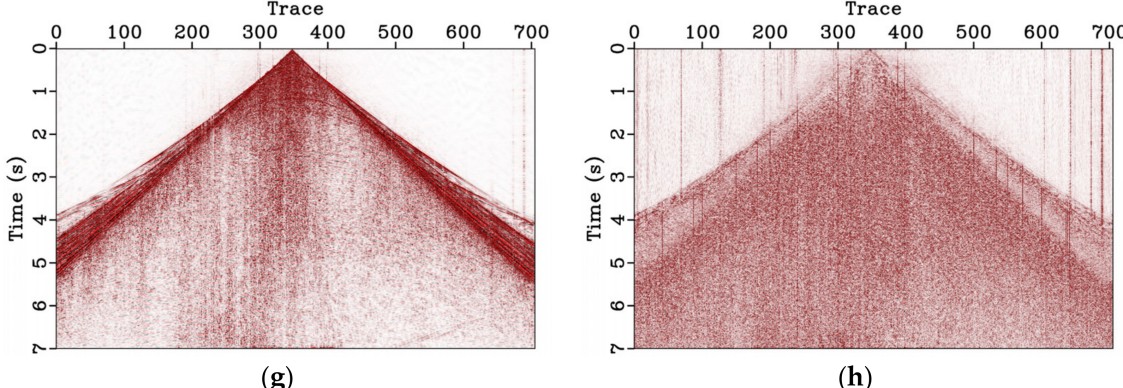

**Figure 15.** Land field data comparison of denoising performance. (**a**) The denoised results using f-x deconvolution. (**b**) Figure (**a**) corresponds to the removed noise. (**c**) The denoised results using the CS method. (**d**) Figure (**c**) corresponds to the removed noise. (**e**) The denoised results using the MF + FX method. (**f**) Figure (**e**) corresponds to the removed noise. (**g**) The denoised results using the proposed method. (**h**) Figure (**g**) corresponds to the removed noise.

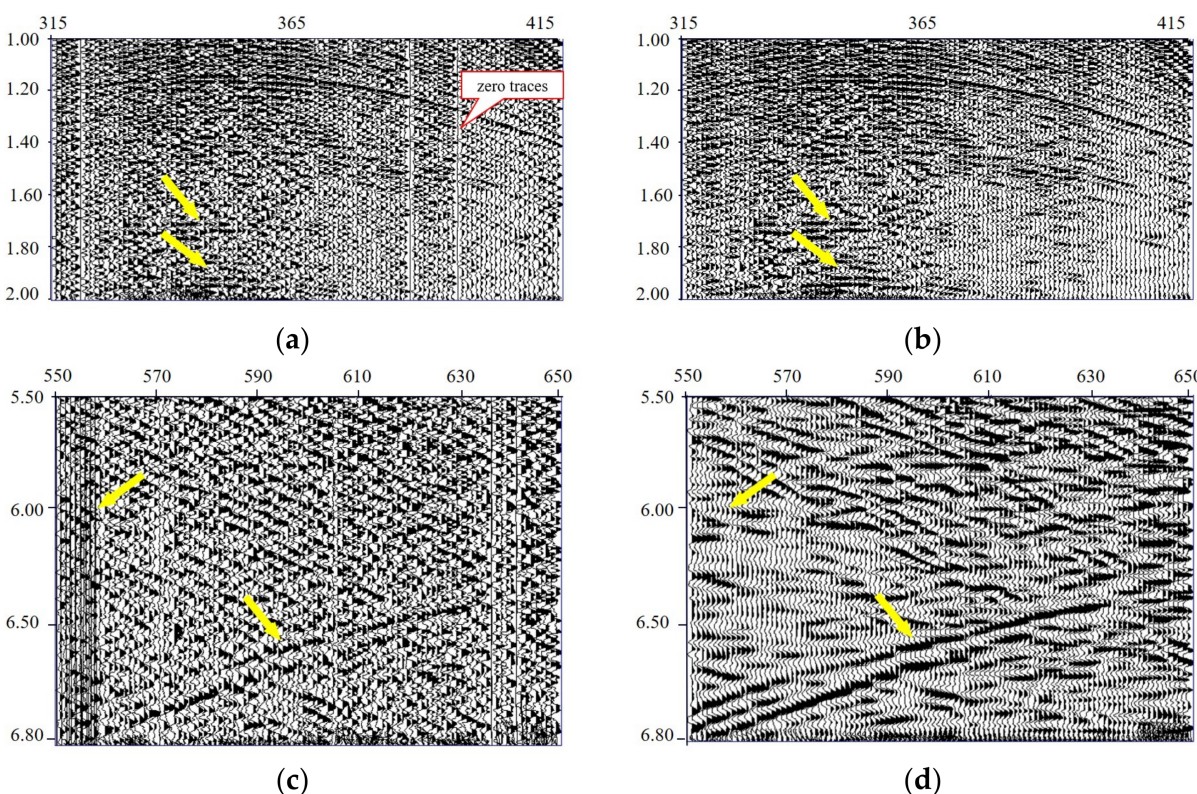

**Figure 16.** Zoomed sections of the field land data. (**a**) True data (up). (**b**) The denoised results using the proposed method. (**c**) True data (down). (**d**) The denoised results using the proposed method.

The fourth example involved worse land field data. The optimized 30% jittered undersampling matrix was used for the denoising test, as shown in Figure 18. It can be seen from Figure 18c,d that this method still demonstrated strong adaptability for low SNR data, and the denoising effect was excellent. This is clearer from the frequency spectra shown in Figure 19; the frequency band of the processed data is more stable.

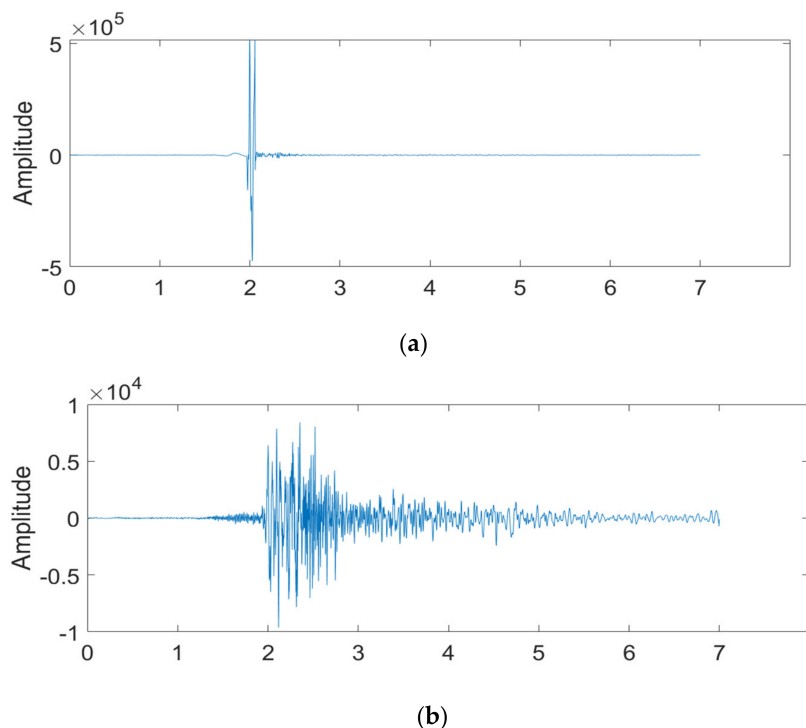

**Figure 17.** 501th trace outlier point analysis. (**a**) Raw data. (**b**) The amplitude after outlier noise suppression.

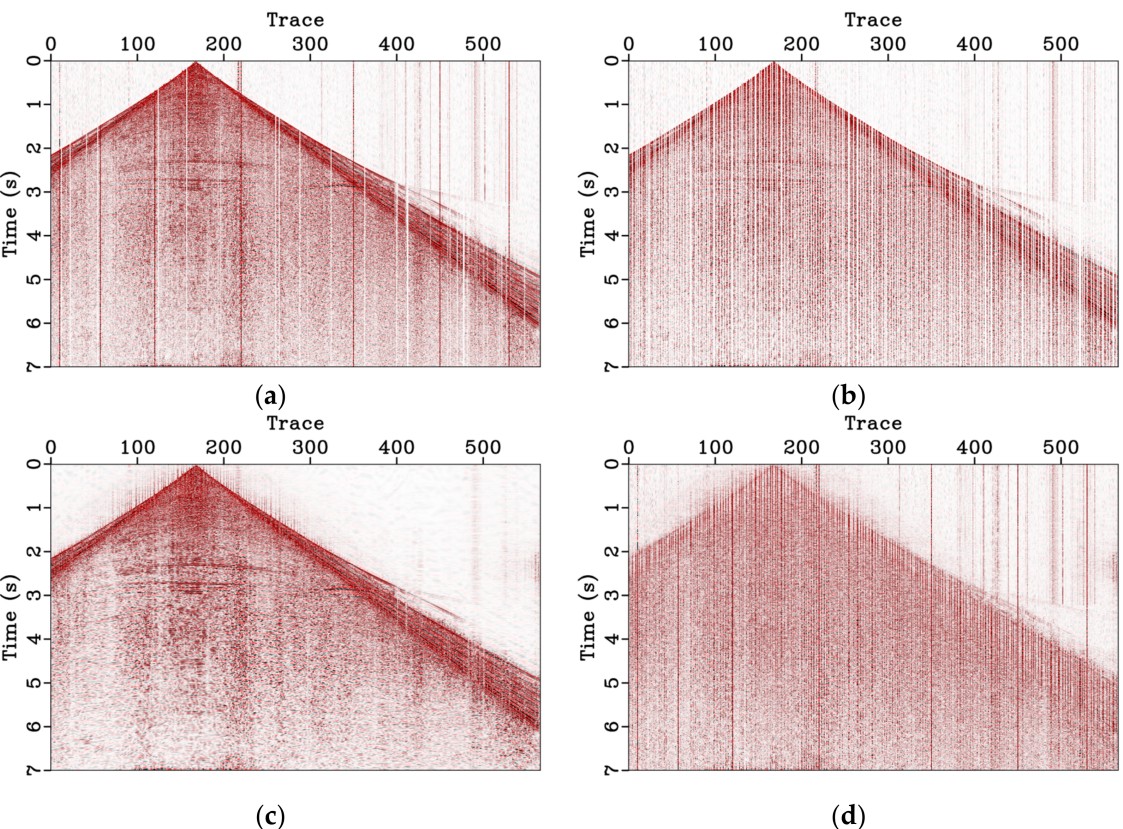

**Figure 18.** Land field data of denoising performance. (**a**) Raw noisy data. (**b**) The 30% jittered undersampling with outlier noise. (**c**) The denoising results. (**d**) Difference sections.

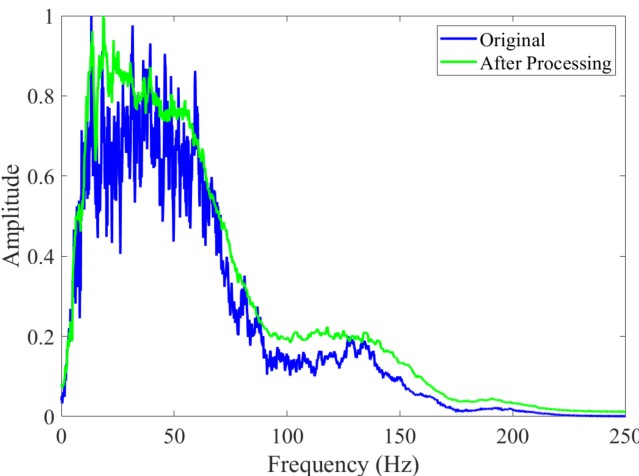

**Figure 19.** Spectrum comparison before and after processing.

## 4. Discussion

The computational costs of the CS sparse inverse algorithm depend largely on the number of iterations. In this study, the synthetic data spatial sampling point used was 200, the time sampling point was 1001, the sampling time was 2 ms, the internal loop was 5, the outer loop was 10 times, and the calculation cost was 11.43 s. For the relatively complex second set of land field data, the dimensions were $567 \times 3501$, the inner loop was 7 times, the outer loop was 20 times, and the calculation cost was 163.54 s. The computational time is nearly identical to the traditional iterative CS denoising method time. In terms of the parameters, the number of internal and external cycles, and the selection of the threshold directly affect the final denoising effect and reconstruction. Compared with the conventional CS denoising algorithm, this method added the outlier value recognition threshold standard $T_o$. The closer the outlier factor $T_o$ value was to 1, the stronger the ability to identify outliers in seismic data.

Gan et al. [38] analyzed the applicability of the iterative threshold (IT) algorithm and the projection-onto-convex-sets (POCS) algorithm in seismic data reconstruction. Both algorithms are common methods used for seismic data reconstruction. However, from the mathematical analysis, the following (Equation (19) is the iteration threshold algorithm formula, and (Equation (20)) is the POCS iterative formula:

$$x_i^{j+1} = T_{\lambda_{ik}}(x_i^j + \Theta_i^T(d - \Phi_{tm}\Psi_i x_{\lambda_i}^j)) \tag{19}$$

$$d_i^{j+1} = d + (I - \Phi_{tm})\Psi_i \cdot T_{\lambda_{ik}}(\Psi_i \hat{d}_i^j) \tag{20}$$

where $x_i^j$ and $d_i^j$ denote the sparse coefficients and the estimated data after $j \times i$ the iteration, respectively. As can be seen from the position of the threshold, the IT algorithm assumes that the global unknown is updated, while the POCS algorithm only sets the missing data as unknown. Compared with the two methods, the IT algorithm is more suitable for data with higher noise levels, while the POCS algorithm has more advantages in protecting the non-missing parts of the data.

In CS theory, the randomness of data is the key to converting a false frequency into a random form, so the randomness of the sampling matrix is very important. Gaussian random sampling has the best incoherence, but the resulting random matrix will cause a problem of large local intervals and data redundancy. Therefore, this research applied the jittered undersampling matrix.

Based on the characteristics of seismic signal components, the degree of sparsity expressed in different sparse transforms was also different. The sparser the information about the sampling signal, the better the reconstruction results. The Fourier transform has a limited ability to convert sparsity [39]. The Radon transform is more suitable for

transforming data with straight-line characteristics, but is not suitable for prestack seismic data [40]. The singular points of data can be well represented by wavelet transforms, but edge data, such as curve features, cannot be optimally represented [41,42]. The K-SVD method, based on the learned dictionary, can constantly update the dictionary to ensure sparsity of the data, but the computational cost of updating this dictionary is large, making this method unsuitable for a wide range of practical applications [43]. The curvelet transform has the characteristics of being multi-scale and multi-directional after several generations of development [44]. It is more suitable as a sparse representation of seismic data, and is widely used in the field of seismic exploration. This study mainly focused on the outlier noise problem in land field data, hence we chose the IT algorithm based on the curvelet transform.

## 5. Conclusions

In this study, we proposed a novel statistics-based mask strategy for CS, in order to remove outlier noise. First, we analyzed the amplitude characteristics with outlier-containing data, based on the standard deviations. According to the standard deviation difference of noise and effective signals, we designed a mask matrix to zero noisy traces. Then, we compared the mask matrix of the outlier trace locations with the jittered undersampling matrix, and replaced the jitter sampling traces that had outlier noise with their nearby unsampled traces. Finally, the curvelet-domain iterative threshold algorithm was used to reconstruct the missing data. The masking-of-outlier-noise strategy showed high computational efficiency because it was not repeatedly computed in the iterative algorithm. Numerical examples for both synthetic and field data demonstrated the feasibility and adaptability of the proposed method.

**Author Contributions:** Conceptualization, W.W. and J.Y.; methodology, W.W. and J.Y.; software, W.W., J.Y. and M.S.; software, W.W. and M.S.; validation, W.W. and J.Y.; formal analysis, W.W. and J.Y.; investigation, W.W.; resources, J.Y., J.H. and M.S.; data curation, W.W. and J.Y.; writing—original draft preparation, W.W. and J.Y.; writing—review and editing, W.W. and J.Y.; visualization, W.W. and J.H.; supervision, W.W. and Z.L.; project administration, J.H. and Z.L.; funding acquisition, J.H., J.Y. and Z.L. All authors have read and agreed to the published version of the manuscript.

**Funding:** This research is supported by the National Key R&D Program of China under contract number 2019YFC0605503C, the major projects of CNPC under contract number ZD2019-183-003, the major projects during the 14th Five-year Plan period under contract number 2021QNLM020001, the National Outstanding Youth Science Foundation under contract number 41922028, and the Funds for Creative Research Groups of China under contract number 41821002. J.Y. is supported by the startup funding (No. 20CX06069A) of Guanghua Scholar at China University of Petroleum (East China), National Natural Science Foundation of China: 42074133, major scientific and technological cooperation projects of petrochina: ZD2019-183-003, the key technology of full node seismic processing: 30200020-21-ZC0607-0021.

**Data Availability Statement:** The synthetic dataset and network models associated with this research can be obtained by contacting the corresponding author.

**Acknowledgments:** This research is supported by the Key Laboratory of deep oil and gas, China University of petroleum (East China). The research was carried out at the National Supercomputer Center in Tianjin, and this research was supported by TianHe Qingsuo Project—special fund project in the field of geoscience. The authors thank the reviewers and editors for their efforts.

**Conflicts of Interest:** The authors declare no conflict of interest.

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
