# Peer review of "Outlier Denoising Using a Novel Statistics-Based Mask Strategy for Compressive Sensing"

_remotesensing, doi:10.3390/rs15020447_

Round 1

Reviewer 1 Report

This paper proses to suppress outlier noise in seismic data using statistic-based mask method. The theory is solid and complete. Numerical tests show the effectiveness of proposed method. I have the following specific comments:  

C01: The Outlier noise article needs to be supplemented.

C02: The evaluation standard of the outlier point in this article is evaluated by standard deviation, have other statistical methods been tested, such as median, mean, etc., or what are the advantages of the standard deviation evaluation method?

C03: The contribution of this paper mainly proposes a mask matrix based on statistical outlier trace recognition, and the mask matrix can be considered to be applied to other interpolation methods to improve computational efficiency.

C04: The horizontal axis of seismogram (such as Figure 15) should keep same unit as in the two dimension profiles.

Author Response

Response to Reviewer 1 Comments

Point 1: The Outlier noise article needs to be supplemented.

Response 1: We have modified the discussion about this point.

Point 2: The evaluation standard of the outlier point in this article is evaluated by standard deviation, have other statistical methods been tested, such as median, mean, etc., or what are the advantages of the standard deviation evaluation method?

Response 2: We have tested different statistical evaluation criteria in the previous research, among which the median value only counts some points in the data, does not cover all the data characteristics, and does not have good judgment; Compared with the median, the mean value counts the characteristics of all the data, but the feature value can amplify the outlier points in the data, so that the anomalous point energy in the data can be better distinguished and eliminated.

Point 3: The contribution of this paper mainly proposes a mask matrix based on statistical outlier trace recognition, and the mask matrix can be considered to be applied to other interpolation methods to improve computational efficiency.

Response 3: Yes, the mask matrix can be a technique to improve data quality, and can achieve good denoising and reconstruction effects based on any reconstruction method.

Point 4: The horizontal axis of seismogram (such as Figure 15) should keep same unit as in the two dimension profiles.

Response 4: We have modified the discussion about this point.

Reviewer 2 Report

In this paper, a mask method based on statistics is developed and applied to the compressed sensing framework to remove outlier noise, which alleviates the signal leakage problem of classical denoising methods. It is worth mentioning that the optimization strategy proposed in this paper uses the idea of problem transformation to converts an outlier denoising problem into a data reconstruction problem, and gets good results. The whole idea of the paper is clear and the data experiment is complete. However, I have the certain number of major concerns that should be solved or at least clarified:

1) What are the benefits of the soft threshold function used in the implementation of the proposed method? Is there, exactly, any improvement over the fixed threshold function? Can we just use the hard threshold function?

2) Two evaluation indexes SNR and PSNR were used to verify the performance of the proposed method in three data sets. However, the comparison test between the proposed algorithm and the classical denoising algorithm in the same scene should be considered. The paper would have been more convincing if the author had added a set of comparative experiments.

3) One SNR or PSNR was used in each data set. In doing so, there are certain limitations to the experimental conclusions that can be reached. Under this concern, I want to encourage the authors to conduct experiments from multiple SNR and PSNR, and draw all the experimental results into a picture. Such experimental results will make the proposed method airtight.

4) There is a problem with the 216 line subheading of the article, which needs to be corrected by the author.

In summary, I don't recommend the publication of this paper in the present form.

Author Response

Response to Reviewer 2 Comments

Point 1: What are the benefits of the soft threshold function used in the implementation of the proposed method? Is there, exactly, any improvement over the fixed threshold function? Can we just use the hard threshold function?

Response 1: The selection of threshold functions is a key issue in iterative threshold algorithms. Both soft and hard thresholds can be used as threshold functions in this paper, but the soft thresholds are chosen because the hard threshold functions are intermittent and will oscillate during reconstruction, resulting in distortion of the imaging contour. The effective signal of the actual data tested in this paper is weak, and the selection of the hard threshold function is very likely to cause signal leakage, so the soft threshold is more suitable for the low SNR data in this paper, and the hyperbolic model and ocean data model can be processed by the hard threshold function.

Point 2: Two evaluation indexes SNR and PSNR were used to verify the performance of the proposed method in three data sets. However, the comparison test between the proposed algorithm and the classical denoising algorithm in the same scene should be considered. The paper would have been more convincing if the author had added a set of comparative experiments.

Response 2: Based on this question raised by experts, we have added a comparative test of the method of median filtering to suppress outlier noise. The difference between the SNR and the PSNR ratio is compared

Point 3: One SNR or PSNR was used in each data set. In doing so, there are certain limitations to the experimental conclusions that can be reached. Under this concern, I want to encourage the authors to conduct experiments from multiple SNR and PSNR, and draw all the experimental results into a picture. Such experimental results will make the proposed method airtight.

Response 3: We have added a comparison of multiple SNR and PSNR according to expert advice.

Point 4: There is a problem with the 216 line subheading of the article, which needs to be corrected by the author.

Response 4: We have modified the discussion about this point.

Reviewer 3 Report

This study proposes a method to eliminate outliers from seismic traces using compressive sensing. Because the authors succeed in demonstrating the utility of the method they propose, and the manuscript is well-organized, this paper should be published after my comments below are appropriately addressed. 

1. The authors acknowledge that many methods have already been proposed to eliminate outliers. To demonstrate the superiority of the method the authors developed, the authors should compare the denoising by the authors' method with other methods, such as Fourier Transform, Radon Transform, K-SVD, and methods outlined in lines 45-65. 

2. The authors employ curvelet as the basis function. I advocate their choice, but it is beneficial for readers to show us a figure to demonstrate the shape of representative curvelet functions. 

Author Response

Response to Reviewer 3 Comments

Point 1: The authors acknowledge that many methods have already been proposed to eliminate outliers. To demonstrate the superiority of the method the authors developed, the authors should compare the denoising by the authors' method with other methods, such as Fourier Transform, Radon Transform, K-SVD, and methods outlined in lines 45-65.

Response 1: Compared with methods such as Fourier transform, the more commonly used contrast method for suppressing outlier noise in the industry is the method of median filtering. We added a number of pictures comparing with the median filtering method in the model test, and used the combination of median filtering and F-X deconvolution to outlier suppression in the terrestrial data model. After comparison, the superiority of the method in this paper can be more seen.

Point 2: The authors employ curvelet as the basis function. I advocate their choice, but it is beneficial for readers to show us a figure to demonstrate the shape of representative curvelet functions.

Response 2: The corresponding picture of the curvelet based function has been added to the paper, which illustrates the characteristic advantages of the multi-scale and multi-directionality.

Round 2

Reviewer 1 Report

Thanks for the revision. The manuscript has been improved dramatically. Only some minor comments:

C01: In equation (18), the parameters should be consistent with equation (16) and (17).

Author Response

Responses for Reviewer 1

1.1: In equation (18), the parameters should be consistent with equation (16) and (17).

Response:  We have modified the problem about this point.

Reviewer 2 Report

Please see the attachment for details.

Author Response

Responses for Reviewer 2

2.1: What are the benefits of the soft threshold function used in the implementation of the proposed method? Is there, exactly, any improvement over the fixed threshold function? Can we just use the hard threshold function?

Response:  For complex land data, the soft threshold function can better avoid signal leakage. The soft threshold function can effectively avoid sudden changes in the process of threshold removal, avoid local jitter in the result after denoising, and the soft threshold is relatively smooth in the processing of curved wave domain coefficients. Hard threshold functions can be used for ocean data, or for data that is gently constructed. Threshold based analysis has been added to the paper.
